# Potential Role of HLA Class I Antigens in the Glycolytic Metabolism and Motility of Melanoma Cells

**DOI:** 10.3390/cancers11091249

**Published:** 2019-08-26

**Authors:** Silvia Peppicelli, Jessica Ruzzolini, Elena Andreucci, Francesca Bianchini, Filippos Kontos, Teppei Yamada, Soldano Ferrone, Lido Calorini

**Affiliations:** 1Department of Experimental and Clinical Biomedical Sciences “Mario Serio”, Section of Experimental Pathology and Oncology, Viale G.B. Morgagni, 50-50134 Florence, Italy; 2Department of Surgery, Massachusetts General Hospital, Harvard Medical School, Boston, MA 02114, USA; 3Istituto Toscano Tumori and Center of Excellence for Research, Transfer and High Education DenoTHE University of Florence, Piazza di San Marco, 4, 50121 Florence, Italy

**Keywords:** HLA class I, glycolytic metabolism, motility, melanoma cells

## Abstract

Besides playing a crucial role in immune surveillance, human leukocyte antigens (HLA) possess numerous non-immune functions involved in cell communication. In the present study, screening of a panel of HLA class I- and HLA class II-specific monoclonal antibodies (mAbs) for their effects on the metabolism of human melanoma cells showed for the first time that the HLA-B,C-specific mAb B1.23.2 reduced the expression level of key glycolytic enzymes, but did not affect that of mitochondrial respiration effectors. As a result, the metabolism of melanoma cells shifted from a Warburg metabolism to a more oxidative phosphorylation. In addition, the HLA-B,C-specific mAb B1.23.2 downregulated the expression of glutamine transporter and glutaminase enzyme participating in the reduction of tricarboxylic acid cycle. The HLA-B,C-specific mAb B1.23.2-mediated reduction in energy production was associated with a reduction of melanoma cell motility. On the whole, the described results suggest that HLA class I antigens, and in particular the gene products of HLA-B and C loci play a role in the motility of melanoma cells by regulating their metabolism.

## 1. Introduction

The activation of signaling pathways that alter metabolism of tumor cells plays a major role in tumor progression. In contrast to normal cells, which rely on mitochondrial oxidative phosphorylation to generate energy needed for cellular processes, most cancer cells keep glycolysis highly activated even under aerobic conditions, a phenomenon termed “Warburg effect” [1,2]. This constitutive feature associated with exogenous supply of glutamine, which serves to replenish tricarboxylic acid cycle, may sustain relentless proliferation of cancer cells. However, this metabolic reprogramming is most often combined with an ineffective blood supply. The resulting hypoxia is per se an inducer of the “anaerobic glycolytic switch”, by which glycolysis is uncoupled from respiration. Glycolysis becomes the primary source of ATP and, combined to an enhanced glucose uptake, produces large quantities of secreted lactate and H^+^, reducing extracellular pH of tumors [3]. Acidity of extracellular microenvironment is associated with an enhancement of the aggressive phenotype of cancer cells and a metabolic adaptation to an oxidative phosphorylation [4,5]. Overall, the metabolic dynamic adaptation of tumor cells in different microenvironments represents one of the hallmarks of malignancy.

Abnormalities in Human Leukocyte Antigen (HLA) class I expression represent an additional hallmark of aggressive cancer cells. They facilitate evasion of cancer cells from T lymphocyte (CTL)- mediated recognition and destruction [6]. In addition to their role in immunity, a “non-immune” function of HLA class I antigen might be considered [7]. As Sir Peter Medawar commented on the “Philosophy of Ignorance” in 1978 [8]: “*We do not know for certain why some cells stick together and others do not, but I do believe that mapping of cell surfaces in terms of the MHC is an important move in the right direction*”. Indeed, HLA class I antigens in humans, like their counterparts in mice play a role in the social organization of cells governing the contact inhibition of cell movement [9]. Furthermore, Major Histocompatibility Complex (MHC) class I antigens are associated with various hormone and growth factor receptors. In particular, mouse MHC class I antigens were found to interact with insulin receptor (IR) [10], and HLA class I antigens with epidermal growth factor [11] and interleukin-2 [12] receptors. Liegler et al., using a fluorescence energy transfer method, studied the proximity between IR and MHC class I antigens and found that a significant energy transfer exists among IR and murine MHC class I (K and D) antigens, indicating a proximity within 10 nm [13]. The functional significance of the structural association of IR and MHC class I antigens is suggested by the finding that a peptide, derived from H-2Dk antigen, enhances the effect of insulin promoting glucose uptake in rat adipocytes [14]. The latter finding implies a potential role MHC class I antigens in the metabolism of cells. This possibility prompted us to investigate whether HLA class I-specific monoclonal antibodies (mAbs) could affect the metabolism of melanoma cells and as a result their functional properties. In this study we have screened HLA-specific mAbs for their effects on the metabolic activity of cultured human melanoma cells and we have found for the first time that the HLA-B,C-specific mAb B1.23.2 inhibits glycolysis. The inhibition of glycolysis mediated by HLA-B,C-specific mAb B1.23.2 also causes a reduction of amoeboid motility of melanoma cells. In our study, we used the metastatic melanoma cell lines A375-M6, WM266-4 and M21 as glycolytic addicted cells; in addition, we used the melanoma cell line FO-1, as cells which do not express HLA class I antigens, and the β2-microglobulin-transfected counterpart (FO-1β2) which re-acquires HLA class I antigen expression following transfection with wild-type β2 microglobulin cDNA.

## 2. Results

### 2.1. Glycolysis Inhibition by the HLA-B,C-specific mAb B1.23.2

Cytofluorographic analysis of A375-M6 melanoma cells stained with the HLA-A-specific mAb LGIII-147.4.1, the HLA-B,C-specific mAb B1.23.2 and the HLA-DR,DQ,DP-specific mAb LGII-612.14 demonstrated a high HLA class I antigen expression and a low HLA class II antigen expression (Figure 1). We also used the HLA-A,B,C-specific mAb (DAKO) as an internal control. The staining is specific, since the mAbs did not stain cultured FO-1 human melanoma cells, which express neither HLA class I nor HLA class II antigens [15]. 

Incubation of A375-M6 melanoma cells with the HLA-B,C-specific mAb B1.23.2 (10 µg/mL) for 24 h at 37 °C reconverted their metabolic profile to an oxidative phosphorylation (Oxphos) metabolism (Figure 2). The expression levels of GLUT1, GLUT3, HK2 and LDHA mRNA were downregulated in melanoma cells incubated with the HLA-B,C-specific mAb B1.23.2 (Figure 2A). In addition, mRNA and protein of M2 splice isoform of pyruvate kinase (PK), a key enzyme promoting a glycolytic metabolism, were downregulated in cells incubated with the HLA-B,C-specific mAb B1.23.2. Lastly, the downregulation of pyruvate dehydrogenase kinase 1 (PDK1), a pyruvate dehydrogenase (PDH)-inhibitory enzyme, and the upregulation of pyruvate dehydrogenase phosphatase 2 (PDP2), a PDH-activating enzyme, confirm a metabolic shift toward Oxphos in the HLA-B,C-specific mAb B1.23.2-treated melanoma cells (Figure 2A). PDH is a mitochondrial multi-enzyme complex that catalyzes the oxidative decarboxylation of pyruvate and plays a major role in the regulation of homeostasis of carbohydrate fuels in mammalian cells. This enzymatic activity is regulated by a phosphorylation/dephosphorylation cycle, and the phosphorylation of PDH by a specific pyruvate dehydrogenase kinase (PDK) results in its inactivation. Thus, the resulting inhibition of PDK1 enhances the use of pyruvate in the tricarboxylic acid cycle, causing increased respiration. Quantitative PCR analysis of HLA-B,C-specific mAb B1.23.2-treated melanoma cells showed specific upregulation of a gene associated with mitochondrial biogenesis and oxidative phosphorylation, the peroxisome proliferator-activated receptor gamma, coactivator 1 alpha (PGC-1α) (Figure 2B). Moreover, the expression of microphthalmia-associated transcription factor (MITF), a well-known PGC-1α transactivator in melanoma cells, was upregulated in cells incubated with HLA-B,C-specific mAb B1.23.2 (Figure 2C). Upon PGC-1α-activation, gene expression that stimulates mitochondrial oxidative metabolism and energy homeostasis are induced. Analysis of monocarboxylate transporters showed an increased level of MCT1 (promoter of lactate influx) associated with an unchanged level of MCT4 (promoter of lactate efflux) in HLA-B,C-specific mAb B1.23.2 -treated melanoma cells (Figure 2C).

MCT1, also known as SLC16A1, belongs to the MCT family and facilitates unidirectional proton-linked transport of L-lactate, pyruvate, β-hydroxybutyrate and acetoacetate across the plasma membrane. In parallel with the mRNA downregulation showed in Figure 2A, the protein expression of GLUT1, GLUT3 and PKM2 was also reduced in cells incubated with HLA-B,C-specific mAb B1.23.2 for 24 h at 37 °C. The latter changes were associated with a decreased activation of AKT, an oncogene which plays a key role in the promotion of glucose metabolism (Figure 2C).

In order to confirm the finding that HLA-B,C-specific mAb B1.23.2-treated melanoma cells changed their metabolic profile, the extracellular acidification rate (ECAR), which reflects the rate of glycolysis, was measured using the Seahorse Analyzer. As shown in Figure 2D, the HLA-B,C-specific mAb B1.23.2 reduced both glycolysis and glycolytic capacity in A375-M6 melanoma cells. However, no change was detected in the oxygen consumption rate (OCR) in A375-M6 treated cells (data not shown). Melanoma cells that were incubated with the HLA-B,C-specific mAb B1.23.2 also displayed a rapid decrease of both K-type mitochondrial glutaminase (GLS1 and GLS2), that catalyzes the hydrolysis of glutamine to glutamate and ammonia, and the alanine, serine, cysteine-preferring transporter 2 (ASCT2), which mediates the uptake of glutamine, an essential amino acid used by proliferating tumor cells (Figure 2E). Uptake of glutamine and subsequent glutaminolysis is critical for the activation of the mTORC1 nutrient-sensing pathway, which regulates cell growth and protein translation in cancer cells. However, no change was detected in cell proliferation after incubation with the HLA-B,C-specific mAb B1.23.2 (Figure 2F).

To prove that the effects we have described were caused by interactions of the HLA-B,C-specific mAb B1.23.2 with the gene products of the HLA-B and C loci and not with unrelated molecules, we tested whether the HLA-B,C-specific mAb B1.23.2 had any effects on the metabolism of FO-1 melanoma cells. The latter cells do not express HLA class I antigens because of a structural mutation in β2m encoding gene [15]. As shown in Figure 2G, a 24 h incubation of FO-1 melanoma cells with the HLA-B,C-specific mAb B1.23.2 caused no detectable changes in the expression level of most of the glycolytic markers analyzed. Furthermore, the HLA-A-specific mAb LGIII-147.4.1 caused no detectable change in the level of glycolytic/oxidative markers in melanoma cells (Figure 2F).

Overall, these results suggest that among the HLA-specific mAbs tested, only the HLA-B,C-specific mAb B1.23.2 inhibits glycolysis and glutamine metabolism, possibly reconverting melanoma cells to a more Oxphos metabolism.

### 2.2. Glycolysis Inhibition by the HLA-B,C-specific mAb B1.23.2 in FO-1 Melanoma Cells with Restored HLA Class I Antigen Expression Mediated by Wild Type β2m Transfection

Additional experiments were performed to corroborate the conclusion that the glycolysis inhibition by the HLA-B,C-specific mAb B1.23.2 is mediated by its interaction with the corresponding antigens. In these experiments the FO-1 melanoma (FO-1neo) cells which do not express HLA class I antigens and the β2-microglobulin-transfected counterpart (FO-1β2) which express HLA class I antigens following transfection with wild type β2-m were used as targets. Cytofluorographic analysis showed that FO-1β2 cells were stained by both HLA class I-specific mAb MO736 (DAKO) and HLA-B,C-specific mAb B1.23.2, while FO-1neo cells were stained by neither mAb (Figure 3A,B).

Comparison of the expression level of mRNAs encoding components of some metabolic pathways in FO-1neo and FO-1β2 cells showed that restoration of HLA class I antigen expression was associated with a metabolic profile characterized by an increased glucose (GLUT1 and 3) and glutamine (GLS1) transporter level and an upregulation of HK2, LDHA, PDK1 and PKM2. PGC1α, the Oxphos marker, displayed a slightly increased level in FO-1β2 cells (Figure 3C). Furthermore, analysis of lactate production by FO-1neo and FO-1β2 cells showed that the latter cells produced an almost 2.5-fold higher level of lactate (Figure 3D). These results indicate that FO-1β2 cells display a preferential use of glycolytic pathway. Analysis of the effect of the HLA-B,C-specific mAb B1.23.2 on the metabolic profile of FO-1neo and FO-1β2 melanoma cells showed that this mAb does not modify level of expression of several metabolic markers in FO-1neo cells, but significantly reduces GLUT3 and LDHA expression (Figure 3E), and lactate production in FO-1β2 melanoma cells (Figure 3F). Additionally, the expression level of PDP2 was upregulated without any detectable change in the expression of PDK1 in FO-1β2 melanoma cells incubated with the HLA-B,C-specific mAb B1.23.2 (Figure 3E). Also the metabolic analysis, using the Seahorse Analyzer, showed that the HLA-B,C-specific mAb B1.23.2 induced reduction of glycolysis and glycolytic capacity in FO-1β2 melanoma cells, but caused no detectable change in the glycolysis of FO-1neo melanoma cells (Figure 3G). These results altogether show that the HLA-B,C-specific mAb B1.23.2 reverts the glycolytic phenotype of FO-1β2 melanoma cells. In analyzing metabolic reconversion of FO-1β2 melanoma cells incubated with the HLA-B,C-specific mAb B1.23.2, the GLS1 downregulation and the additional PGC1α upregulation are noteworthy (Figure 3H). In summary, the HLA-B,C-specific mAb B1.23.2 has similar effects on the metabolism of FO-1β2 cells and A375-M6 melanoma cells, since in both cells it inhibits glycolysis and glutamine metabolism converting them to a more oxidative phenotype.

### 2.3. Inhibition by the HLA-B,C-specific mAb B1.23.2 of Melanoma Cell Motility

The bioenergetic profile, and in particular, aerobic glycolysis adopted by cancer cells including proliferating melanoma cells, is known to influence cellular motility as well as actin-dependent cytoskeletal dynamics. Therefore we have tested whether inhibition of the glycolytic metabolism caused by the HLA-B,C-specific mAb B1.23.2 had any effect on melanoma cell motility. 

2-deoxyglucose (2-DG), a classical glycolytic inhibitor, used at a concentration known to inhibit melanoma cell motility was used as a positive control. Using the wound assay, we have shown that 2-DG causes a dose-dependent inhibition of both A375-M6 (Figure 4A) and FO-1β2 (Figure 4B) melanoma cell motility. The HLA-B,C-specific mAb B1.23.2 reduces the motility of A375-M6 melanoma cells (Figure 4C); this effect is mediated by the interaction of HLA-B,C-specific antibodies with the corresponding antigens, since mAb B1.23.2 had no detectable effect on the motility of the FO-1 melanoma cells, which do not express HLA class I antigens (Figure 4D). The role of HLA class I antigens in the motility of FO-1 cells is also suggested by the significantly higher ability of FO-1β2 cells than of FO-1neo cells to close the wound in the scratch assay (Figure 4E). The HLA-B,C-specific mAb B1.23.2 abrogates almost completely this enhanced activity (Figure 4F). Figure 5 shows that also the metastatic melanoma cell lines WM266-4 and M21 which express HLA class I antigens (Figure 5A) display a downregulated glycolytic metabolism (Figure 5B) and a reduced ability to close the wound in the scratch assay when they are incubated with the HLA-B,C-specific mAb B1.23.2 (Figure 5C).

As cell migration is regulated by continuous cytoskeleton remodeling, we compared the effect of the glycolytic inhibitor 2-DG and the HLA-B,C-specific mAb B1.23.2 on cytoskeletal organization, using Phalloidin, a bicyclic peptide belonging to a family of toxins isolated from the deadly *Amanita phalloides* “death cap” mushroom, to selectively label F-actin. We found that in A375-M6 melanoma cells treated with 2-DG, F-actin filaments disassembled, whereas in untreated cells F-actin was normally patterned (Figure 6A). To go inside the transduction pathway of movement inhibited by 2-DG, we evaluated the positivity of melanoma cells for Paxillin, a protein expressed by focal adhesions of cells. Paxillin is a signal transduction adaptor able to bind several proteins, including protein tyrosine kinases, such as Src and focal adhesion kinase, structural protein such as vinculin and actopaxin and regulators of actin organization, such as COOL/PIX and PKL/GIT [16]. Level of Paxillin phosphorylation in A375-M6 melanoma cells treated with 2-DG was dramatically reduced (Figure 6B). Therefore, we analyzed the Phalloidin and Paxillin phosphorylation staining in A375-M6 and FO-1β2 melanoma cells following a 24 h incubation with the HLA-B,C-specific mAb B1.23.2. We found that also this mAb, like 2-DG, was able to significantly reduce cytoskeletal organization and stress fiber formation, as indicated by Phalloidin staining (Figure 6C) and phospho-Paxillin (Figure 6D).

## 3. Discussion

Aerobic glycolysis, also known as “Warburg effect”, appears to be one of the hallmarks of cancer cells. Indeed, cancer cells consume at least ten times more glucose than normal cells [2], a phenomenon visualized in tumor-bearing patients using 18F-deoxyglucose positron emission tomography (FDG–PET) imaging [17]. 

This study shows for the first time that the HLA-B,C-specific mAb B1.23.2 is able to reprogram the glycolysis addiction of melanoma cells. This conclusion is supported by the results of experiments performed with several melanoma cell lines. They include the HLA class I antigen expressing metastatic melanoma cell lines A375-M6, WM266-4 and M21; the FO-1 melanoma cell line which does not express HLA class I antigens, and its counterpart on which HLA class I antigen expression is restored by transfection with β2m transfection. 

Among the HLA-specific mAbs tested, only the HLA-B,C-specific mAb B1.23.2 was able to shift melanoma cells from Warburg metabolism to a more Oxphos. This metabolic reprogramming reflects the downregulation of most key glycolytic enzymes, and the upregulation of mitochondrial respiration markers confirmed by Seahorse analysis. PGC1α, which mediates mitochondrial biogenesis and respiration, and its driver MITF, are upregulated in HLA-B,C-specific mAb B1.23.2-treated A375-M6 and FO-1β2 melanoma cells. In accordance with the elevation of PGC1α, MCT1, an additional Oxphos marker, was over-expressed. MCT1 promotes lactate influx which contributes to provide alternative sources of energy [4,5]. In addition, PDK1 depletion, although present only in HLA-B,C-specific mAb B1.23.2-treated A375-M6 melanoma cells, together with an increase of PDP2, might have a role in triggering a cellular phenotype associated with an enhanced oxygen consumption rate [18]. Xenografts into nude mice of melanoma cells with PDK1 downregulation displayed reduced growth compared with control cells [19]. The additional reduction in glutaminase enzyme (GLS1,2) and glutamine transporter (ASCT2) induced by the HLA-B,C-specific mAb B1.23.2 in melanoma cells, may reduce many of the building blocks derived from tri-carboxylic acid cycle (TCA) intermediates that need to be replenished in a process called anapleurosis. It was found that GLS works as a rate-limiting enzyme for glutamine utilization in proliferating T cells and leukemic cells [20]. 

The results we have described are not unique of HLA class I antigens, since other components of the immune response such as B7-H3, a member of the B7 family of ligands, have been reported to modulate glycolysis. In fact, B7-H3 overexpression increases the glycolytic pathway of triple-negative breast cancer cells, whereas knockdown of B7-H3 decreases glycolysis [21]. Furthermore, Lim et al. have demonstrated that the reprogramming of glucose uptake and lactate production by breast cancer cells mediated by B7-H3 depends on HIF-1α stabilization [22].

The link between aerobic glycolysis and regulation of the actin-dependent cytoskeletal dynamics has been studied widely [23,24,25]. Shiraishi et al. demonstrated that motility of human prostate and breast cancer cells is energetically dependent on aerobic glycolysis, and mitochondrial derived ATP was found insufficient to compensate glycolytic pathway inhibition [26]. In addition, migration of normal cells, such as activated vascular smooth muscle cells (VSMC), was found to be dependent on aerobic glycolysis [26] and the anti-migratory property of indirubin-3′-monoxime may partly depend on impairment of glycolysis via a compromised STAT3/HK2 signaling axis. Overall, these results prompted us to investigate whether the metabolic shift induced by the HLA-B,C-specific mAb B1.23.2 might have an impact on the motility of melanoma cells. Our results demonstrate that the HLA-B,C-specific mAb B1.23.2 can significantly reduce the motility and Paxillin expression in both A375-M6 and FO-1β2 melanoma cells. Further, when we used 2-DG, a glycolytic inhibitor, motility of our melanoma cells was reduced. 2-DG is a potent inhibitor of hexokinase (HK), a crucial enzyme of glycolytic pathway, once it is phosphorylated by HK itself to 2-deoxy-D-glucose-6-phosphate. A mechanism underlying the reduction in motility mediated by the HLA-B,C-specific mAb B1.23.2 may be related to the inhibition of lactate accumulation. Lactate was found to be able to promote tumor cell migration [27]. 

## 4. Materials and Methods 

### 4.1. Cell Lines and Culture Conditions

The melanoma cell line A375-M6, isolated in our laboratory from a lung metastasis in a SCID bg/bg mouse i.v. injected with A375 human melanoma cell line [27], obtained from American Type Culture Collection (ATCC, Rockville, MD, USA); the melanoma cell line FO-1, the FO-1 cell lines transfected with pSV2neo alone (FO-1 neo) or with human wild type β2m gene and pSV2neo (FO-1β2), the human melanoma cell lines WM266-4 (from ATCC) and M21 (kindly provided by Antony Montgomery, The Scripps Research Institute, La Jolla, CA, USA) were cultured in Dulbecco’s Modified Eagle Medium high glucose (DMEM 4500, EuroClone, Figino, MI, Italy) supplemented with 10% fetal bovine serum (FBS, Boerhinger Mannheim, Mannheim, Germany) and 2mM L-glutamine (EuroClone, MI, Italy) at 37 °C in a humidified atmosphere containing 90% air and 10% CO_2_. 

Cells were harvested from subconfluent cultures by incubation with a trypsin-EDTA solution (EuroClone, MI, Italy), and propagated every three days. Viability of cells was determined by trypan blue exclusion test. Cultures were periodically monitored for mycoplasma contamination using Chen’s fluorochrome test [28]. 

### 4.2. Monoclonal and Polyclonal Antibodies

The mouse mAb LGIII-147.4.1 which recognizes an epitope expressed on the gene products of HLA-A locus [29], mAb B1.23.2 which recognizes an epitope expressed on the gene products of HLA-B and C loci [30] and the mAb LGII-612.14 which recognizes an epitope expressed on the b chains of HLA-DR, DQ and DP loci [31] were developed and characterized as described. mAbs were purified from ascitic fluid by affinity chromatography on a Protein G column. The activity and purity of mAb preparations were monitored by binding assays with the corresponding antigens and by SDS-PAGE.

Alexa Fluor 488-conjugated polyclonal anti-mouse IgG antibody was purchased from Invitrogen (Carlsbad, CA, USA).

### 4.3. Cell Treatments

Cells were treated for 24 h at 37 °C with HLA-specific mAbs, at the concentration of 2 or 10 μg/mL. In some experiments cells were treated for 24 h at 37 °C with 2-Deoxy-D-glucose (2DG) at concentrations ranging between 0.25 and 40 mM.

### 4.4. Proliferation Assay

Cell proliferation was evaluated using CellTrace™ CFSE Cell Proliferation Kit (Life Technologies, Monza, Italy) as previously described [32]. Melanoma cells were incubated for 20 min at 37 °C with the dye at the concentration of 5 μM; then the cells were cultured alone or in the presence of the HLA-B,C-specific mAb B1.23.2. At different timepoints, the cells were detached, fixed (paraformaldehyde 5%) and analyzed by flow cytometry. The fluorescence value obtained was analyzed by ModFit software to estimate the proliferation index. 

### 4.5. Flow Cytometer 

Cells were harvested by using Accutase (Euroclone, Milan, Italy), collected in flow cytometer tubes (2 × 10^5^ cells/tube), centrifuged at 1200× *g* for 5 min and incubated for 1 h at 4 °C with the HLA-specific mAbs at the final concentration of 5 μg/mL. Cells were then washed in PBS and incubated 1 h in the dark at 4 °C with anti-mouse antibody conjugated with Alexa Fluor 488 (Invitrogen, Carlsbad, CA, USA) diluted 1:1000 in PBS. Samples were washed in PBS and resuspended in 500 µL PBS to proceed with the analysis at BD FACSCanto (BD Biosciences, Franklin Lakes, NJ, USA). The flow cytometer was calibrated using cells incubated with secondary antibody only. For each sample, 1 × 10^4^ events were analyzed.

### 4.6. Western Blotting Analysis

Cells were washed with ice cold PBS containing 1 mM Na_4_VO_3_, and lysed in 100 μL of cell RIPA lysis buffer (Merk Millipore, Vimodrone, MI, Italy) containing PMSF (Sigma-Aldrich, Saint Louis, MO, USA), sodium orthovanadate (Sigma-Aldrich) and protease inhibitor cocktail (Calbiochem). Immunoblot was performed as described previously in [32]. Aliquots of supernatants containing equal amounts of protein in Laemmli buffer were separated on Bolt^®^ Bis-Tris Plus gels 4–12% precast polyacrylamide gels (Life Technologies, Monza, Italy). Fractionated proteins were transferred from the gel to a PVDF nitrocellulose membrane using an electroblotting apparatus (Bio-Rad, Segrate, MI, Italy). Blots were stained with Ponceau red to ensure equal loading and complete transfer of proteins, then they were blocked for 1 h, at room temperature, with Odyssey blocking buffer (Dasit Science, Cornaredo, MI, Italy). Subsequently, the membrane was probed at 4 °C overnight with primary antibodies diluted in a solution of 1:1 Odyssey blocking buffer/T-PBS buffer. The primary antibodies were: rabbit anti-MCT-1, rabbit anti-MCT-4, mouse anti-MITF (Santa Cruz Biotechnology, Santa Cruz, California), rabbit anti-AKT, rabbit anti p-AKT, rabbit anti-GLUT1, rabbit anti-GLUT-3 and rabbit anti-PKM2 (Cell signaling Technology, Danvers, MA, USA). The membrane was washed in T-PBS buffer, incubated for 1 h at room temperature with goat anti-rabbit IgG Alexa Flour 680 antibodies (Invitrogen, Monza, Italy), and then visualized by an Odyssey Infrared Imaging System (LI-COR^®^ Bioscience). Mouse anti-α-tubulin monoclonal antibody (Sigma, Saint Louis, MO, USA) was used to assess equal amount of protein loaded in each lane.

### 4.7. RNA Isolation and Quantitative PCR (qPCR)

Total RNA was isolated from cells by using TRI Reagent (Sigma). The amount and purity of RNA were determined spectrophotometrically. cDNAwas obtained by incubating 2 μg of total RNA with 4 U/μL of M-MLV reverse transcriptase (Promega, San Luis Obispo, CA, USA) according to the manufacturer’s instructions. 

Quantitative real time PCR (qPCR) was performed as reported in [27] using the GoTaq^®^ Probe Systems (Promega). The qPCR analysis was carried out in triplicate using an Applied Biosystems 7500 Sequence Detector with the default PCR setting: 40 cycles at 95 °C for 15 s, 60 °C for 60 s. mRNA was quantified with the ∆∆Ct method as described [33]. mRNA levels were normalized to 18S as an endogenous control. The primer sequences used are listed in Table 1.

### 4.8. Lactate Assay

Lactate level in the media of cell cultures was measured using the Lactate Assay kit (Source Bioscience Life Sciences) [28]. Samples were prepared in 50 μL/well with Lactate Assay Buffer in a 96-well plate. Reaction mix (50 μL) containing 46 μL lactate assay buffer, 2 μL probe and 2 μL enzyme mix was added to each well and the plate was incubated for 30 min at room temperature, protected from light. Lactate reacted with the enzyme mix to generate a product, which interacted with lactate probe to produce fluorescence. Fluorescence (Ex/Em = 535/590 nm) was measured in a microplate reader.

### 4.9. Seahorse Analysis 

The oxygen consumption rate (OCR, pmolesO_2_ consumed/min) and the extracellular acidification rate (ECAR, mpH/min) were determined using the Seahorse XF96 Extracellular Flux Analyzer (Seahorse Bioscience, Billerica, MA, USA). Two kits were used. The Agilent Seahorse XF Cell Mito Stress Test Kit (Agilent Technologies, Santa Clara, CA, USA) quantitates the OCR of cells to measure parameters related to mitochondrial function. The Seahorse XF Glycolysis Stress Test Kit (Agilent Technologies) measures preferentially the glycolytic function in cells.

Cells were counted and seeded (3 × 10^4^ cells/well) in XF96 Seahorse^®^ microplates precoated with poly-D-lysin (ThermoFisher Scientific, Waltham, MA, USA). In order to assess OCR or ECAR, cells were suspended in XF Assay Medium supplemented with 1 mM pyruvate, 2 mM glutamine and 10 mM glucose (all from EuroClone, Paington, UK) for OCR or 1 mM glutamine for ECAR. Cells were left to adhere for a minimum of 30 min at 37 °C. Then 100 μL of XF Assay Medium were added to each well. The plate was left to equilibrate for 10 min in a CO_2_-free incubator before being transferred to the Seahorse XF96 analyzer. The pre-hydrated cartridge was filled with the indicated compounds and calibrated for 30 min in the Seahorse Analyzer. All the experiments were performed at 37 °C. Normalization to protein content was performed after each experiment. The Seahorse XF Report Generator automatically calculated the parameters from Wave data that have been exported to Excel.

### 4.10. Wound Healing Assay

Cell migration was evaluated by an *in vitro* wound healing assay as described [34]. Cells were grown in 35 mm dishes until they reached confluence; cell monolayer was wounded with a sterile 200 mL pipette tip, washed with PBS and incubated in 1% FBS culture medium. Wound was analyzed following a 24-h incubation and photographed using phase contrast microscopy.

### 4.11. Immunofluorescence

Cells were grown on glass coverslips in 6-well plates for 24 h. Culture medium was removed, cells washed with PBS, fixed 30 min at 4 °C with 3.7% paraformaldehyde and permeabilized for 15 min with PBS 0.1% Tryton X-100 at room temperature. Cells were then incubated 1 h at room temperature with blocking buffer (0.1% Tryton X-100 and 5.5% horse serum PBS). Following a 1 h-incubation at room temperature with rhodamine-phalloidin (Invitrogen Molecular Probes R415) or anti-phospho-paxillin antibody (Cell Signaling 2541) diluted 1:100 in PBS, Alexa Fluor 488-conjugated anti-mouse IgG antibodies (Invitrogen, Carlsbad, CA, USA) were added to glass coverslips and incubation was continued for 45 min at room temperature in the dark. Subsequently, nuclei were stained with DAPI dye (4′,6-diamidino-2-phenylindole; Thermo Fisher Scientific, Waltham, MA, USA) for 20 min at room temperature in the dark. Cells were then dried, mounted onto glass slides, and examined with confocal microscopy using a Nikon Eclipse TE2000-U (Nikon, Tokyo, Japan). A single composite image was obtained by superimposition of 20 optical sections for each sample analyzed.

### 4.12. Statistical Analysis

The experiments were performed at least three times for a reliable application of statistics. Densitometric data are expressed as means ± standard errors of the mean (SEM) of the results obtained from three independent representative experiments. The unpaired Student’s *t*-test was used to evaluate pair-wise differences, with *p* < 0.05 being considered significant. 

## 5. Conclusions

In conclusion, the results we have presented indicate that besides playing a crucial role in immune response, HLA class I antigens are crucial in the metabolism of melanoma cells, sustaining Warburg effect as well as glutamine metabolism. Both metabolic pathways contribute to cell motility, which is critically involved in tumor cell dissemination. If so, HLA-B,C antigens represent a new target to affect cancer cell metabolism and motility.

## Figures and Tables

**Figure 1 cancers-11-01249-f001:**
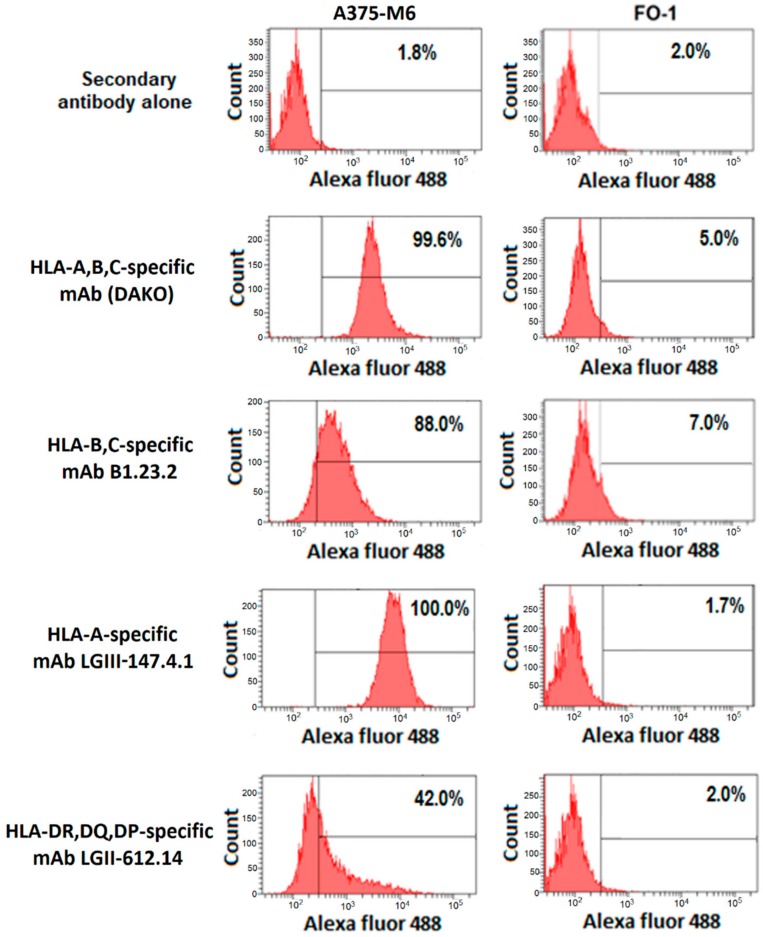
HLA class I and II antigen expression on melanoma cells. A375-M6 (left panel) and FO-1 (right panel) melanoma cells were stained with HLA-specific mAbs and analyzed with a flow cytometer. Numbers indicate the % of stained cells.

**Figure 2 cancers-11-01249-f002:**
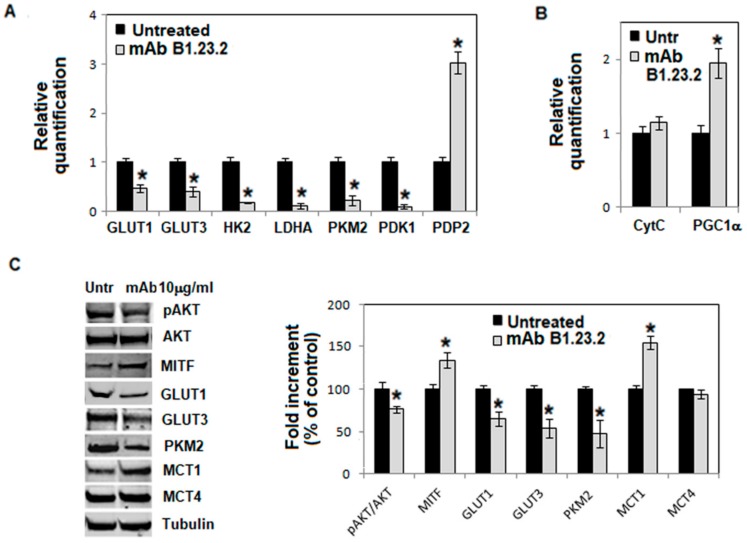
Effect of the HLA-B,C-specific mAb B1.23.2 on the metabolism of A375-M6 melanoma cells. Analysis of mRNA expression by quantitative real-time PCR of genes involved in glycolysis (**A**), oxidative phosphorylation (**B**) and glutamine metabolism (**E**) in A375-M6 melanoma cells treated with the HLA-B,C-specific mAb B1.23.2 (10 μg/mL) for 24 h at 37 °C. Western blot analysis of pAKT, AKT, MITF, GLUT-1, GLUT-3, PKM2, MCT-1 and MCT-4 expression in melanoma cells treated with the HLA-B,C-specific mAb B1.23.2 (10 μg/mL) for 24 h at 37 °C. Each band in the Western blot was quantified by densitometric analysis and the corresponding histogram was constructed by normalizing the density of each band to that of α-tubulin. Representative Western blot panels are shown on the left (**C**). Glycolysis and glycolytic capacity, extracted from glycolysis stress assay results obtained by analyzing with the Seahorse Analyzer A375-M6 melanoma cells treated with the HLA-B,C-specific mAb B1.23.2 (10 μg/mL) for 24 h at 37 °C (**D**). Proliferation index, evaluated by CFSE staining, of A375-M6 cells grown in the presence of the HLA-B,C-specific mAb B1.23.2 at 37 °C for up to 72 h (**F**). Analysis of mRNA expression by quantitative real-time PCR of genes involved in glycolysis (left panel), oxidative phosphorylation (middle panel), and glutamine metabolism (right panel) in A375-M6 melanoma cells treated with the HLA-A-specific mAb LGIII-147.4.1 (10 μg/mL) for 24 h at 37 °C (**G**). Quantitative real-time PCR of genes involved in cell metabolism of FO-1 melanoma cells treated with the HLA-B,C-specific mAb B1.23.2 (10 μg/mL) for 24 h at 37 °C (**H**). Values presented are the mean ± SEM of the results obtained in three independent experiments. * *p* < 0.05.

**Figure 3 cancers-11-01249-f003:**
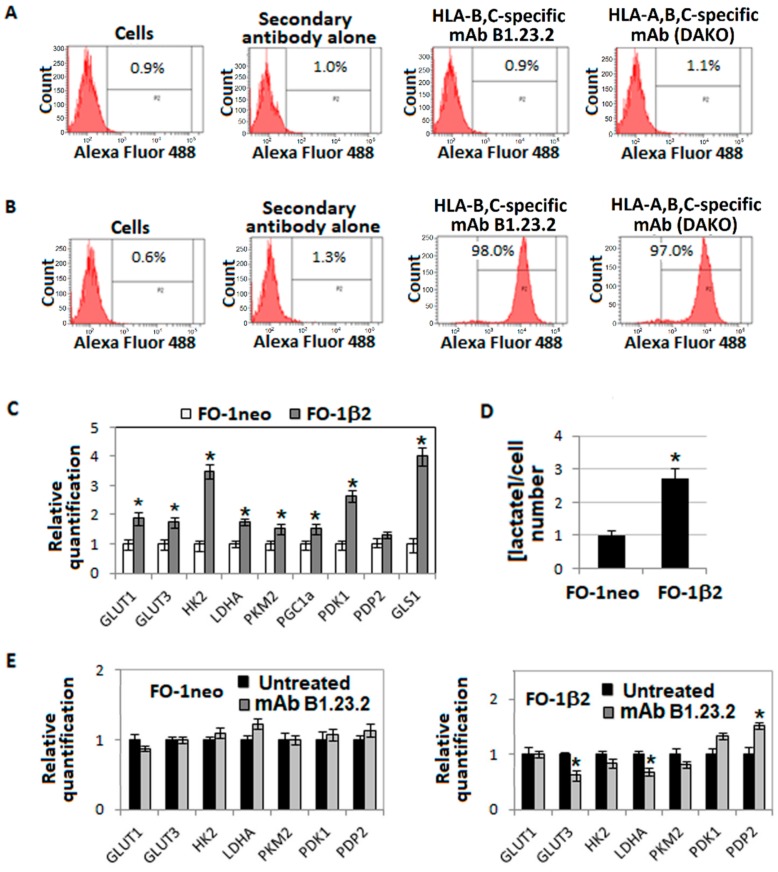
Effect of the HLA-B,C-specific mAb B1.23.2 on the metabolism of FO-1neo/FO-1β2 model of melanoma cells. FO-1neo (**A**) and FO-1β2 (**B**) melanoma cells were stained with the HLA-B,C-specific mAb B1.23.2 and analyzed with a flow cytometer. Representative plots are shown in the panels. Evaluation by quantitative real-time PCR of genes involved in metabolism in FO-1neo or FO-1β2 cells (**C**). Lactate released by FO-1neo or FO-1β2 melanoma cells corrected for number of cells (**D**). Evaluation by quantitative real-time PCR of genes involved in glycolytic metabolism (**E**) or in oxidative metabolism (**H**) in FO-1neo or FO-1β2 cells treated with the HLA-B,C-specific mAb B1.23.2 for 24 h at 37 °C. Lactate released by FO-1neo or FO-1β2 melanoma cells treated with the HLA-B,C-specific mAb B1.23.2 for 24 h at 37 °C corrected for number of cells (**F**). Glycolysis and glycolytic capacity extracted from glycolysis stress assay results obtained using the Seahorse Analyzer of FO-1neo and FO-1β2 melanoma cells treated with the HLA-B,C-specific mAb B1.23.2 (10 μg/mL) for 24 h at 37 °C (**G**). Values presented are the mean ± SEM of the results obtained in three independent experiments. * *p* < 0.05.

**Figure 4 cancers-11-01249-f004:**
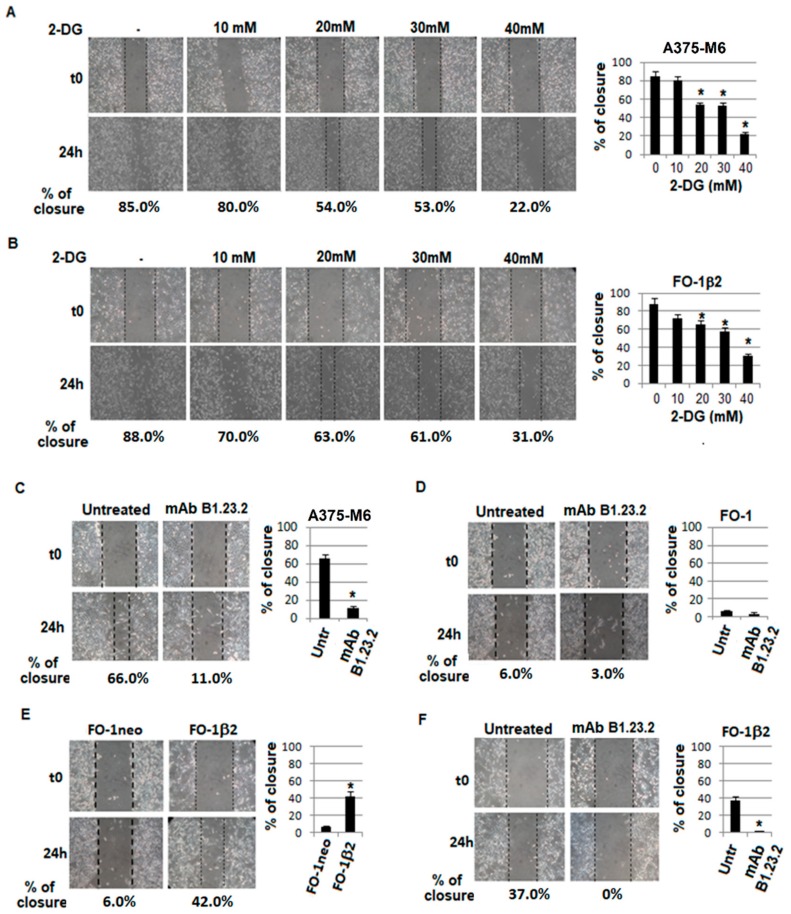
Effect of 2DG and HLA-B,C-specific mAb B1.23.2 on cell motility. Effect of 2-DG on: motility of A375-M6 cells (**A**) and FO-1β2 cells (**B**), evaluated by scratch wound healing assay. A wound healing assay was used also to evaluate the migration of A375-M6 (**C**) and FO-1 (**D**) cells treated with the HLA-B,C-specific mAb B1.23.2 (10 μg/mL) for 24 h at 37 °C; differential migration of FO-1neo and FO-1β2 cells (**E**) and migration of FO-1β2 cells after treatment with the HLA-B,C-specific mAb B1.23.2 (10 μg/mL) for 24 h at 37 °C (**F**). Values presented are the mean ± SEM of the results obtained in three independent experiments. * *p* < 0.05.

**Figure 5 cancers-11-01249-f005:**
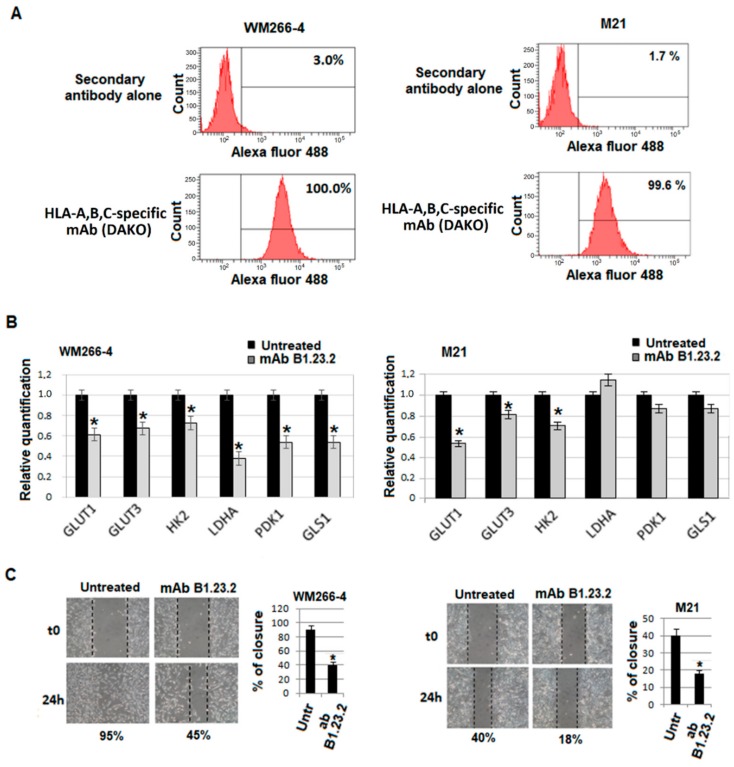
HLA-A,B,C expression on WM266-4 and M21 melanoma cells. WM266-4 (left panel) and M21 (right panel) melanoma cells were stained with the HLA-A,B,C-specific mAb purchased from DAKO and analyzed with a flow cytometer (**A**). Analysis of mRNA expression by quantitative real-time PCR of genes involved in metabolism in WM266-4 (left panel) and M21 (right panel) melanoma cells treated with the HLA-B,C-specific mAb B1.23.2 (10 μg/mL) for 24 h at 37 °C (**B**). Motility of WM266-4 and M21 melanoma cells in the presence of the HLA-B,C-specific mAb B1.23.2 (10 μg/mL) for 24 h at 37 °C (**C**). Values presented are the mean ± SEM of the results obtained in three independent experiments. * *p* < 0.05.

**Figure 6 cancers-11-01249-f006:**
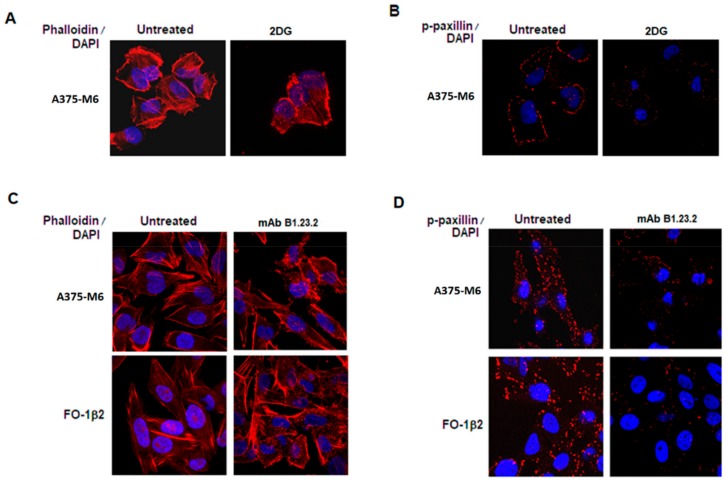
Staining of F-actin with rhodamine–Phalloidin (**A**) and intracellular localization of phospho-paxillin (**B**) in A375-M6 treated with 20 mM 2DG for 24 h at 37 °C. Staining of F-actin with rhodamine–Phalloidin (**C**) and intracellular localization of phospho-Paxillin (**D**) in A375-M6 and FO-1 cells treated with the HLA-B,C-specific mAb B1.23.2 (10 μg/mL) for 24 h at 37 °C. Scale bar = 10 μm.

**Table 1 cancers-11-01249-t001:** Primer sequences used for PCR.

Gene	FW	RV
GLUT1	5′-CGGGCCAAGAGTGTGCTAAA-3′	5′-TGACGATACCGGAGCCAATG-3′
GLUT3	5′-CGAACTTCCTAGTCGGATTG-3′	5′-AGGAGGCACGACTTAGACAT-3′
LDHA	5′-AGCCCGATTCCGTTACCT-3′	5′-CACCAGCAACATTCATTCCA-3′
PKM2	5′-CAGAGGCTGCCATCTACCAC-3′	5′-CCAGACTTGGTGAGGACGAT-3′
PDK1	5′-CCAAGACCTCGTGTTGAGACC-3′	5′-AATACAGCTTCAGGTCTCCTTGG-3′
HK2	5′-CAAAGTGACAGTGGGTGTGG-3′	5′-GCCAGGTCCTTCACTGTCTC-3′
18s	5′-CGCCGCTAGAGGTGAAATTCT-3′	5′-CGAACCTCCGACTTTCGTTCT-3′
PGC1a	5′-GGGAAAGTGAGCGATTAGTTGAG-3′	5′-CATGTAGAATTGGCAGGTGGAA-3′
PDP2	5′-ACCACCTCCGTGTCTATTGG-3′	5′-CCAGCGAGATGTCAGAATCC-3′
CytC	5′-TTGCACTTACACCGGTACTTAAGC-3′	5′-ACGTCCCCACTCTCTAAGTCCAA-3′
GLS1	5′-TGCTACCTGTCTCCATGGCTT-3′	5′-CTTAGATGGCACCTCCTTTGG-3′
GLS2	5′-TGCCTATAGTGGCGATGTCTCA-3′	5′-GTTCCATATCCATGGCTGACAA-3′
ASCT2	5′-GGTGGCTGGCAAGATCGT-3′	5′-CCAAGGCGGGCAAAGAG-3′

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
