# Peer review of "Potential Role of HLA Class I Antigens in the Glycolytic Metabolism and Motility of Melanoma Cells"

_cancers, 2019, doi:10.3390/cancers11091249_

Round 1

Reviewer 1 Report

The study by Peppicelli et al. describes that melanoma cell metabolism is modified by ligation of MHC class I proteins with mAb B1.23.2. Moreover, FO-1ß2 cells displaying restored MHC class I expression are more glycolytic than FO-1neo cells. These results are novel and interesting and suggest a new potential function of MHC class I proteins besides antigen-presentation.

Finally, the study links glycolytic capacity to migratory capacity of melanoma cells.

I wonder whether the authors have tried to modulate the metabolism of rapidly growing (glycolytic) normal cells with mAb B1.23.2?

Minor points

·     Due to the importance of the results obtained with mAb B1.23.2, the authors should describe in more detail its purification. Did the authors use several batches of this mAb or only 1 batch?

·     Line 44. The sense of the following sentence is unclear: HLA class I antigens represent an additional hallmark of aggressive cancer cells. Do the authors mean: The presence of HLA class I antigens represent an additional hallmark of aggressive cancer cells?

·     Line 79. The authors probably meant: The expression levels of GLUT1, GLUT3, HK2 and … instead of,  The expression levels of GLUT1, 3, HK2 and …

·     In several Figure legends (Fig. 2-4) the concentration of mAb is expressed as 10 mg/mL when actually are 10 µg/mL.

·     References #2 and #17 are the same.

Author Response

Point 1. I wonder whether the authors have tried to modulate the metabolism of rapidly growing (glycolytic) normal cells with mAb B1.23.2?
This is a very interesting observation; however, normal cells are usually not glycolytic addicted and use respiration when oxygen tension is enough to sustain viability, only some precursors cells may use glycolysis or normal cells undergoing an ischemia attack, when they change their metabolic attitude toward anaerobic glycolysis. We think that this aspect deserve a more profound study which might be take into consideration in a next future.

Minor points
· Due to the importance of the results obtained with mAb B1.23.2, the authors should describe in more detail its purification. Did the authors use several batches of this mAb or only 1 batch?
We used different batches of the mAb B1.23.2. For the detailed description of the mAb B1.23.2 see the reference #30.

Line 44. The sense of the following sentence is unclear: HLA class I antigens represent an additional hallmark of aggressive cancer cells. Do the authors mean: The presence of HLA class I antigens represent an additional hallmark of aggressive cancer cells?”
We rephrased the sentence according to Rev.1 (see line 44).

Line 79. The authors probably meant: The expression levels of GLUT1, GLUT3, HK2 and ... instead of, The expression levels of GLUT1, 3, HK2 and ...
We changed the text (see line 79).

In several Figure legends (Fig. 2-4) the concentration of mAb is expressed as 10 mg/mL when actually are 10 µg/mL.
We fixed the errors in Figure Legends

References #2 and #17 are the same. We Fixed the error in “references”section.

Reviewer 2 Report

Dear Editor,

I reviewed the manuscript by Peppicelli et al., entitled ‘’  Potential role of HLA class I antigens in the glycolytic metabolism and motility of melanoma cells ‘’

The manuscript is interesting , but the data of the study need to be confirmed in several melanoma cell lines. Also, the authors needs to investigate whether the reduction melanoma motality is associated with cell growth inhibition. Thus, some additional experiments are needed. For example analysis of cell viability using MTT assay. Also, The analysis of the cell cycle by flow cytometry using propodeum Iodide (PI).

Author Response

The manuscript is interesting , but the data of the study need to be confirmed in several melanoma cell lines.

We added figure 5 with some results obtained using M21 and WM266-4 melanoma cell lines treated with the mAb B1.23.2. We confirmed that the mAb B1.23.2 downregulates glycolytic markers (see PCR of GLUT1, GLUT3, HK2, LDHA, PDK1) and motility of M21 and WM266-4 cells.

Also, the authors need to investigate whether the reduction melanoma motility is associated with cell growth inhibition. Thus, some additional experiments are needed. For example analysis of cell viability using MTT assay. Also, The analysis of the cell cycle by flow cytometry using propodeum Iodide (PI).

We analyzed cell proliferation using CellTrace™ CFSE Cell Proliferation Kit after cell treatment with the B1.23.2 mAb and we didn’t find any difference (We added the result in figure 2 F). In addition, cell count using Trypan blue exclusion test did not show any difference between treated and untreated cells.

Reduction in glycolytic rate induced by HLA B1.23.2 mAb is found critical for cell movement, but not for cell proliferation possibly meaning that this metabolic inhibition affects more profoundly ATP generation than production of intermediates for cell proliferation.

Round 2

Reviewer 2 Report

Dear Editor,

The manuscript is improved and can be published in the present form

Many thanks